# Scheduling and Sizing of Campus Microgrid Considering Demand Response and Economic Analysis

**DOI:** 10.3390/s22166150

**Published:** 2022-08-17

**Authors:** Li Bin, Muhammad Shahzad, Haseeb Javed, Hafiz Abdul Muqeet, Muhammad Naveed Akhter, Rehan Liaqat, Muhammad Majid Hussain

**Affiliations:** 1School of Electrical and Electronics Engineering, North China Electric Power University, Beijing 100096, China; 2Department of Electrical Engineering, Muhammad Nawaz Sharif University of Engineering and Technology, Multan 60000, Pakistan; 3Department of Electrical Engineering Technology, Punjab Tianjin University of Technology Lahore, Lahore 54770, Pakistan; 4Department of Electrical Engineering, University of Engineering and Technology (RCET), Lahore 54890, Pakistan; 5Department of Electrical Engineering, Government College University Faisalabad, Faisalabad 38000, Pakistan; 6Department of Electrical and Electronic Engineering, University of South Wales, Pontypirdd CF37 1DL, UK

**Keywords:** smart grid, batteries, campus microgrid, prosumer market, distributed generation, energy storage system

## Abstract

Current energy systems face multiple problems related to inflation in energy prices, reduction of fossil fuels, and greenhouse gas emissions which are disturbing the comfort zone of energy consumers and the affordability of power for large commercial customers. These kinds of problems can be alleviated with the help of optimal planning of demand response policies and with distributed generators in the distribution system. The objective of this article is to give a strategic proposition of an energy management system for a campus microgrid (µG) to minimize the operating costs and to increase the self-consuming energy of the green distributed generators (DGs). To this end, a real-time based campus is considered that currently takes provision of its loads from the utility grid only. According to the proposed given scenario, it will contain solar panels and a wind turbine as non-dispatchable DGs while a diesel generator is considered as a dispatchable DG. It also incorporates an energy storage system with optimal sizing of BESS to tackle the multiple disturbances that arise from solar radiation. The resultant problem of linear mathematics was simulated and plotted in MATLAB with mixed-integer linear programming. Simulation results show that the proposed given model of energy management (EMS) minimizes the grid electricity costs by 668.8 CC/day ($) which is 36.6% of savings for the campus microgrid. The economic prognosis for the campus to give an optimum result for the UET Taxila, Campus was also analyzed. The general effect of a medium-sized solar PV installation on carbon emissions and energy consumption costs was also determined. The substantial environmental and economic benefits compared to the present situation have prompted the campus owners to invest in the DGs and to install large-scale energy storage.

## 1. Introduction

Power systems have been undergoing many problems which include greenhouse gas emission (GHG), inflating consumption cost, complex network overloading, etc. The conventional grid is not able to solve these issues but the emerging smart microgrid system with distributed generators (DGs) equipped with an intelligent distribution system and energy storage systems has the capacity to mitigate problems related to the scheduling of resources by implementing demand response solutions. A campus microgrid (µG) on the other hand, consists of storage systems, onsite DGs, and organized loads [1]. It may additionally operate both in an island mode or in grid-connected mode [2]. The developing trend of microgrids provides an effective solution to monitor the system intelligently and has the ability of self-recovery, persuasive control, and high-tech control with the help of overall sensor installation [3,4,5]. The smart and efficient grid offers diverse possibilities for renewable energy implementation for prosumer µGs by integrating energy management (EMSs) systems. Various kinds of energy management systems need secure interaction between prosumer and conventional grid to operate the control devices intelligently [6,7,8]. However, the distribution network includes a group of µGs wherein every µG acts as a self-governing distribution node, consequently, µGs consist of onsite DGs, storage systems, and DR programs which may play a significant role in minimizing network overloading and electricity cost. The aforementioned benefits have been much reported for multiple µGs with excessive loads. University campus buildings are one of the excessive load µGs with shortfall under the loads of customers due to the changing nature of the electrical loads [9,10,11].

However, these type of institutional buildings can distribute their surplus electricity to the grid community with the presence of onsite electricity generation resources while serving as a general prosumer [12,13,14]. In the same way, in extreme load scenarios, when campus DGs and battery storage are insufficient to meet the load requirement, they may additionally import the necessary electricity from the grid. The actual contribution of such campus µGs in operations of the grid not only minimizes the energy functioning cost but also assists in the supply network [15,16,17] Microgrid operators also supply a proposition on many incentive-based and price-based multiple DR programs to appeal to the large-scale customers in the energy markets. Energy managing solutions are used with the existing resources to assure the best optimal dispatch to meet load demand at a decreased price and thereby ensuring their active participation in supporting the grid operations [18,19,20].

The main objective of this research focuses on EMS development and an economical improvement for a campus µG energy consumption with reduced operational costs which has onsite DGs and a storage facility. The smart micogrid solution increases energy self-consumption while lowering electricity costs and system load operating costs during peak hours, which is addressed by a smart optimal solution for a campus microgrid. The solution allows for the optimal scheduling of multiple energy resources, which lowers the operational costs of the energy resources. Battery degradation costs and the ideal campus microgrid sizing are also considered in order to enhance the mathematical modeling of campus µG. For our U.E.T Taxila, Campus Microgrid, a techno-economic analysis was undertaken with the scheduling of the energy resources under various case studies in order to lower operational costs. The given proposition of EMS can effectively manage the bidirectional power flow optimally among utility networks and µG, and can optimally plan the charging/discharging patterns accordingly to reduce the cost of energy. For general analysis, the real load of an actual Taxila, Campus was taken into consideration. Currently, the considered campus µG has an electrical grid network connection with the nearby distribution company called Islamabad Electric and Supply Company (IESCO) which also includes an external backup diesel generator and wind power as an external source. The environmental and economical effects of PV-based energy storage and energy production in this proposition are also investigated.

## 2. Literature Review

A µG system comprised of solar panels, combined heat and power (CHP), diesel engines, and storage battery for various cities of (Pakistan) was simulated in ref. [21] by HOMER Pro microgrid software. The main purpose was to minimize the electricity generation cost, total net cost, and yearly GHG emissions while improving the grid sales and to increase the yearly waste heat recovery methods of thermal units that transfer the additional waste heat into extra energy. The analysis was executed in two types of modes: islanded mode and grid-connected mode. It was studied and analyzed that every type of city has an optimum special objective function, however, the competent authority makes an optimum decision to choose an optimal city according to their objective. The overall analysis shows that Lahore city has the minimum GHG emissions (1000.314 tons) annually while the city Quetta has the largest grid sales (8,322,368 kWh) annually among various cities.

On the other hand, Rehman et al. [22] proposed a smart campus microgrid model for customers having a national grid, PV units, batteries, and flexible loads while maintaining grid reliability and sustainability. The feasibility of the given system was analyzed for the Levelized Cost (LCOE) of Energy and the net cost with the help of HOMER Pro software. The electricity cost having no grid outage was calculated (0.135$ per kWh). The best possible setup for the household microgrid was calculated to be solar PV capacity with 2 kW, battery energy storage with 1200 Ah, and a power converter with 1 kW. From this kind of setup, the system maintenance, and operational costs, capital costs, and the replacement cost were to be $6522, $7610, and $2833.

In ref. [23], the authors developed a scheduling framework for the PV-Storage-based campus microgrid considering battery degradation cost and battery running cost. The comparative analysis was conducted and proposed with the current literature of the MILP model (Mixed-Integer Linear Programming). The proposed system included a solar PV plant and an energy storage system. The proposed model minimized the electricity cost, battery degradation cost, and the peak-demand violation penalty. In addition, it addressed the two main issues; the optimum use of batteries with the help of RTCS (Real-Time Control Schemes) and to lessen the solar irradiance forecasting error. To manage the SOC (State-of-Charge), the FAM (Flexible Assignment Method) technique was implemented and its costs were minimized from 36,286,470 (KRW) to 34,354,895 (KRW).

Furkan et al. [24] presented an optimal hybrid system for the Ali-Garh University, India. The proposed system consists of a PV (Photo-Voltaic) system, BESS system, and diesel operated generators which maintain the power level as a backup in the campus microgrid. Homer Pro software is implemented as an optimization tool for the microgrid. Results show that the investment cost of this system (PV + BESS) is 1419.6 million Indian rupees, 64.2% more than the current system (864.2 million Indian rupees). However, the Investment cost of Solar PV + Grid is 494.92 million Indian rupees, 57.2% lesser than the current system which is the optimal solution.

Zhang et al. [25] were given the testbed project for the campus microgrid of Georgia Institute of Technology. The proposition of that paper was given to 400 net meters and a group of 200 commercial buildings and it was performed on OpenDSS software. A huge amount of data for the distributed system was controlled by the latest data management system. The DR (Demand Response) based strategies were implemented that aimed to improve the interaction between commercial buildings and the grid. It also studied the expansion of electricity generation planning as future research.

Miguel et al. [26] presented a test-bed smart ERESMA grid concept that was installed at the De Vega Zana campus, Spain. This ERESMA grid facility consists of a two solar PV system in which both systems are incorporated by 22 PV mono linked silicon modules with peak power calculated as 11.6 kWp. Results are compiled by the calculation of 19 technical–economical KPI (Key Performance Indicators) of which some are mentioned here. Capitol cost of this facility or test-bed is 2831 € per kW_DC_, the Levelized cost is 383.5 € per MWh, payback time is 12.4 years, and OPR (Overall Performance Ratio) is 92.4%, etc. Though, here the economic feasibility of the system was unnoticed in the proposed model. In ref. [27], the authors developed a scheduling framework for the PV-storage-based microgrid considering battery degradation cost and battery running cost. The comparative analysis was conducted and proposed with the current literature of the MILP model (Mixed-Integer Linear Programming).

The proposed formulation of the system represented in Figure 1 consists of prosumer µG, electric grid, and EMS. The campus µG contains several kinds of energy storage technologies, loads, and three different distributed energy resources (diesel generators, solar PV, and wind turbine).

In addition, the authors in ref. [23] also addressed the two main issues; the optimum use of batteries for the campus microgrids with the help of RTCS (Real-Time Control Schemes) and to lessen the solar irradiance forecasting error. To manage the SOC (State-of-Charge), the FAM (Flexible Assignment Method) technique was implemented and its costs were minimized from 36,286,470 (KRW) to 34,354,895 (KRW).

Authors of ref. [28], were given a load reduction model of a utility grid keeping in view the grid availability for the residential customers by implementing the linear programming simulated in the MATLAB software. The affordability of PV-storage systems was addressed in this paper with the consideration of multiple hours of load shedding by online optimization methods or techniques. Multiple situations of load shedding were investigated, and the results were analyzed for 8 h of load shedding that could save almost 1000 kWh and for an average household almost 1200 kWh. Moreover, the authors observed that a 4 h random load shedding scenario minimizes the monthly energy consumption cost by up to 16%. However, the author in ref. [29] devised a linear programming technique, in which optimization of RERs and BESS are appropriately scheduled. This proposed model is solved in MATLAB and includes grid, solar, and ESS after scheduling. Results show that optimal price-based scheduling reduces the cost from 532.96 to 516.08, 63%, reducing carbon footprints.

Chen huo et al. [30] presented a multi-approach proposition to protect the campus microgrid for energy deficit in which the μ-grid synchronous phasor approach and power protection laboratory approach are considered. These two approaches are analyzed here on research based for the Oregon Corvallis campus (OSU). In this paper, the OSU campus microgrid consists of smart meters, 2-Solar PV arrays, and systems to focus on energy management. By using this approach, it analyzes the system according to the voltage regulations and load variations at peak times. However, Kaisar et al. [31] provided some research on the flow of the water stream to the combined heat and power generator with the control, distributed, and monitoring mechanism. In that paper, the main objective was to get warnings before system failures which were accessible by monitoring the changeable loads. A faulty system was prevented by smart sensors, that record system data and send it to users to show them possible failures. The Campus microgrid consists of a 15 kW PV system and two/65 kW CHP generators for the McNeese University, USA. Finally, it focuses on IoT devices aimed for energy savings and minimizing the production cost.

A structural design for the smart energy microgrid was also proposed by Gonzalez in [32] for the Eindhoven University of Technology, Netherlands. It analyzed the existing campus grid and aimed to convert it into a smart campus microgrid. The author suggests that smart campus technologies are needed to increase the efficiency of a power system. Results are analyzed on the optimized generic algorithm in MATLAB which illustrates that almost 400 kWh energy production is obtained from the charging/discharging techniques but a more optimized solution is needed to reduce the operational cost for the system. These objectives are analyzed in ref. [33] that aims to contribute an analysis for the campus microgrid and further implements multiple MG in a system that optimally controls together the Chonnam Campus Microgrid, Korea. It consists of 500 kW ESS, PV, load controllers, and a power load bank. It trades the energy under the P2P trading mechanism. Results show that it maximizes the performance of every microgrid that is interlinked with multiple MG and EP-Agents but a DSS (Decision Support System) can also be incorporated to further improve the system performance with reduced operational cost keeping in focus.

Zhang et al. [25] were given the testbed project for the campus microgrid of the Georgia Institute of Technology. The proposition of this paper was presented for 400 net meters and a group of 200 commercial buildings and performed on the OpenDSS software. A huge amount of data for the distributed system was controlled by the latest data management system. The DR (Demand Response) based strategies were implemented which aimed to improve the interaction between commercial buildings and the grid. It also studied the expansion of electricity generation planning as future research. Federico et al. [34] proposed an electrical circuit model to schedule optimally energy production to save actual energy. This proposed scenario reduces the difference between temperatures calculated from the circuit. This testbed is for the Savona Smart Campus. Results compiled from the temperature profiles show that for 11,000 kWh, it saves almost 1400 kWh with this electrical circuit model.

Hoe and Coe [35] proposed a BESS scheduling model that solves the issues of demand response for the campus microgrid. The given model minimized the uncertainties in demand response deployment and operational system cost [29,36]. The total costs were minimized from 85.1$ to 42.7$ with DR involvement. Storage technologies have helped with various applications in managing different kind of microgrids. Among many applications, off-grid system applications [24], energy arbitrage [37], distribution system deferral [38,39,40], frequency regulation [41], demand-side management [42], peak reductions [43], and power system reliability [44,45], etc., are the key contributions for energy storage systems. Several technologies of ESS exist such as flywheel, compressed air, BESS, etc. The optimum charging/discharging patterns of the energy storage system can additionally improve the efficacy and life of the battery. From all of these advantages, BESS technologies are considered here in this paper among with Li-ion technology.

Several related works cited above, especially on the power management structure of the microgrid, considered optimal planning, ESS and PV. Different researchers here studied the integration of ESS into a microgrid while examining the feasibility of solar PV, but some other researchers only focused on reducing the cost of PV and scheduling for ESS, LCOE with simultaneous consideration of energy exchange with utilities, PV uncertainties, battery degradation costs, and demand response. This work investigates previously mentioned research areas and presents a concise comprehensive model of the energy management structure of a campus microgrid with the help of optimal scheduling and planning of an economical and optimum energy storage system with reduced electricity consumption.

The main contributions of this paper are as follows.

A smart optimal solution is presented for a campus microgrid to reduce the operational cost of the energy resources by which multiple energy resources are optimally scheduled in such a way as to increase self-consumption and to reduce the electricity cost and system load running costs during peak hours, which is the main factor for the Taxila campus microgrid. Additionally, to improve the mathematical modelling of campus µG, battery degradation costs are considered and the optimal sizing of the campus microgrid is focused on. Similarly, to reduce the operational cost, a techno-economic analysis was conducted with scheduling of the energy resources in different case scenarios, while giving an optimum solution for our U.E.T Taxila, Campus Microgrid.

The following sections constitute the remainder of the paper. In Section 3, the suggested system’s architecture and formulation are given. The suggested model’s results and discussion are presented in Section 4, and the conclusions of this article are summarized in Section 5.

## 3. Methodology

### 3.1. Proposed Conceptual Framework

The given proposition of EMS employed in the prosumer network that normally takes the data of weather, load demand, unit prices, the ESS early status, and input data is taken as the associated parameters, to search a best possible optimum solution that can satisfy the demand with the resources available without violation of operation and design limitations. The best result is then directed to the system control scheduler to schedule the available resources of the system. It also provides a facility to store many significant parameters, which can be used to produce many benefits for future purposes. Real-time market database and a prosumer database stores the electricity exchange data, prosumer load data, and price data. This proposed model is presented in the Figure 2.

### 3.2. Problem Methodology

Keeping in view the service life of the BESS system, the proposed mathematical model is modeled as a linear constraint optimization problem that is able to lessen the operating costs of µG prosumers. The system constraints are associated with several given model components that are in general mentioned below.

### 3.3. Objective Function

This proposed model has the objective to decrease the operational *cost* (*J*) of a µG, that includes the cost of energy exchange, the cost of wind turbine, diesel generator cost, and degradation cost of energy storages (2)–(5). Formula (1) gives the sum of different types of costs. The battery life depends on several factors, which consist of the used number of cycles, capital costs, and the overall system capacity, as shown in the Formulas (4)–(6) whereas storage is represented by $\eta_{ch}$, $\eta_{dch}$, $P_{t}^{ch}$ and $P_{t}^{ch}\;$ [40] which are respectively represented by Formula (5):(1)$$C_{T}=J=min\sum_{t=1}^{24}(cost_{t}^{E}+cost_{t}^{DG}+cost_{t}^{ESS}+cost_{t}^{WT}+cost_{t}^{BESS})\;$$
whereas,
(2)$$cost_{t}^{E}=P_{\left(t\right)}^{G}\gamma_{t}$$
(3)$$cost_{t}^{DG}=\alpha T_{Gen}+\beta p_{\left(t\right)}^{DG}$$
(4)$$cost_{t}^{WT}=S_{c}.P_{rated}\left(\$\right)\;$$
(5)$$cost_{t}^{ESS}=\left(\frac{C_{cost}}{n\times CT\times 2}\right)\times \left(\eta_{c}p_{\left(t\right)}^{ch}+\frac{p_{\left(t\right)}^{dch}}{\eta_{dch}}\right)$$
(6)$$cost_{t}^{BESS}=S_{BESS}\left(C_{i}^{ESS}+C_{m}^{ESS}t_{om}\right)$$
(7)$$P_{\left(battery\right)}=\eta_{\left(ch\right)}P^{ch}{}_{\left(t\right)}-\frac{p_{\left(t\right)}^{dch}}{\eta_{\left(dch\right)}\;}$$
where $cost_{t}^{E}$,$\;cost_{t}^{WT}$,$\;cost_{t}^{ESS}$, $cost_{t}^{DG}$ are exchange cost of energy, wind turbine cost, degradation cost of battery, and diesel generator cost at the time interval *t*. The campus reserved the general time of use (TOU) tariff connection from the electricity supply company named IESCO. Throughout any interval *t*, the energy trade with utility grid and the energy unit price are represented by $P_{\left(t\right)}^{G}\;$ and $\gamma_{t}$ respectively. $cost_{t}^{DG}$ is found by using the rated capacity of the diesel generator (*T_G_ =* 600 kW), fuel intercept curve (α  = 0.0166 L/h per kW), fuel curve slope (β = 0.277 L/h per kW) and the overall generated power ($P_{\left(t\right)}^{DG}$) from DG [46]. The regular charging power efficiency, charging power, discharging power efficiency, and energy storage discharging power are characterized by $\eta_{\left(ch\right)}$, $p_{\left(t\right)}^{ch}$, $\eta_{\left(dch\right)}$, and $p_{\left(t\right)}^{dch}$ respectively and the battery net power ($P_{\left(battery\right)}$) is signified in Formula (7).

### 3.4. Load Balancing Equality Constraint

The equality constraint of the load basically represents the equilibrium constraint between supply and demand. In order to attain this equilibrium, the Formula (8) must be fulfilled and satisfied. Among them, $P_{t}^{pv}$ and $P_{t}^{l}$ are respectively the output of solar photovoltaic power generation in kW and the prosumer load demand [7].
(8)$$P_{\left(t\right)}^{G}+P_{\left(t\right)}^{PV}+P_{\left(t\right)}^{Battery}+P_{\left(t\right)}^{DG}+P_{\left(t\right)}^{WT}+P_{\left(t\right)}^{BESS}=P_{\left(t\right)}^{total}$$

### 3.5. Energy Storage System Constraints

ESS is a not to be ignored element in the energy management system, because it assists in the supply of electrical load in the event of a grid failure [47]. Since the ESS normally cannot be easily charged or discharged immediately, its power limit was included in the limits (9)–(13). In any interval *t* “*BSOC_t_* “, the battery state of charge in the ESS depends on its earlier state *BSOC*_(*t*−1)_, that is merged in Formula (14). In order to remove the ESS overload and whole discharging, the BSOC maximum and minimum limits are respectively represented by *BSOC_max_* and *BSOC_min_* in Formula (15). As shown in Formula (16), the battery’s state of charge (*BSOC_T_*) at the end of a day is equivalent to its initial battery state (*BSOC*_0_) occurring at the start of the day [7].
(9)$$\frac{BSOC_{t-1}-BSOC_{max}}{100}C^{es}\leq P_{\left(t\right)}^{Battery}$$
(10)$$P_{\left(t\right)}^{Battery}\leq \frac{BSOC_{\left(t-1\right)}-BSOC_{\left(min\right)}}{100}C^{es}$$
(11)$$0\leq \eta_{\left(ch\right)}P_{\left(t\right)}^{ch}\leq Y_{t}^{ch}P_{\left(ch,\;\;max\right)}^{Battery}$$
(12)$$0\leq \frac{P_{\left(t\right)}^{dch}}{\eta_{\left(dch\right)}}\leq Y_{t}^{dch}P_{\left(dch,max\right)}^{Battery}$$
(13)$$Y_{t}^{ch}+Y_{t}^{dch}\leq 1\forall t\;$$
(14)$$BSOC_{\left(t\right)}=BSOC_{\left(t-1\right)}-\frac{100\times \eta_{\left(dch\right)}P_{\left(t\right)}^{dch}}{C^{es}}-\frac{100\times P_{\left(t\right)}^{dch}}{C^{es}\eta_{\left(dch\right)}}$$
(15)$$BSOC_{\left(min\right)}\leq BSOC_{\left(t\right)}\leq BSOC_{\left(max\right)}\;$$
(16)$$BSOC_{\left(T\right)}=BSOC_{\left(0\right)}\;$$

The battery power output $P_{t}^{bat}$ has been added to the equality constraint given in Formula (8) to effectively schedule the energy participation in EMS. The positive and negative values of $P_{t}^{bat}$ represent ESS charging and discharging, respectively. In any interval “*t*”, the ESS charging and discharging are signified by the two integer variables $\mu_{t}^{ch}$ and $\mu_{t}^{dch}$, respectively. To best avoid the BESS charging and discharging problem at similar timings, the given binary variables available in Formulas (11)–(13) cannot be “1” at similar times. For any of these variables, a value equal to “1” indicates the activation mode.

The output power gradient of the energy storage is given below:(17)$$\left|P_{\left(t\right)}^{Battery}-P_{\left(t+1\right)}^{Battery}\right|\leq \Delta P^{Battery}$$

### 3.6. Optimal Sizing of BESS

In order to increase the economic benefit, an optimal sizing methodology is adopted to provide a peak load shaving strategy for an electrical consumer by enhancing the BESS lifetime [48].
(18)$$P_{\left(t\right)}^{DG}=P_{fi}^{wt}\left(t\right)+P_{fi}^{PV}\left(t\right)\;$$
(19)$$P_{\left(t\right)}^{DG}=P_{\left(t\right)}^{dl}-P_{\left(t\right)}^{DG}\;$$
(20)$$P^{D}{}_{\left(t\right)}=P^{dl}{}_{\left(t\right)}-P^{DG}{}_{\left(t\right)};\;\;P^{dl}{}_{\left(t\right)}>P^{DG}{}_{\left(t\right)}\;$$

In Formula (18)–(20), $P_{\left(t\right)}^{DG}$ is the distributed generation power at time interval *t*, $P_{\left(t\right)}^{dl}$ is the power of the actual demand of the load at time interval *t*, and $P^{D}{}_{\left(t\right)}$ is the deficiency of power at time *t*.
(21)$$E^{D}=\sum_{t=1}^{t=24}P^{D}{}_{\left(t\right)}\;$$
(22)$$S_{BESS}=\frac{E^{D}}{\left(1-\rho \right)}\;$$
(23)$$Cost_{t}^{BESS}=S_{BESS}\left(c_{i}^{ESS}+c_{ESSm}k_{om}\right)\;$$

In Formulas (21)–(23),$\;E^{D}$ is the energy providing by the battery (kWh), $S_{BESS}$ is the battery energy storage system size rated in (kWh), $Cost_{t}^{BESS}$ is battery energy storage system cost rated in ($/kWh), $c_{ESSm}$ is the energy storage system maintenance cost rated in ($/kWh), $c_{i}^{ESS}$ is the ESS installation cost rated in ($/kWh), $k_{om}$ is the maintenance factor, and ρ is the battery state of charge.

### 3.7. DG and Grid Constraints

Since utilities incorporate their system components based on the load demand, they constantly sign peak demand contracts with consumers. Any request beyond the requirements of this contract has the consequence of fines or loss in power connection. In the same way, diesel generators cannot meet loads exceeding their rated capacity. The supply of power limitations is considered for the diesel generator and the grid connection by Formulas (24) and (25).
(24)$$P_{\left(min\right)}^{G}\leq P_{\left(t\right)}^{G}\leq P_{\left(max\right)}^{G}$$
(25)$$P_{\left(min\right)}^{DG}\leq P_{\left(t\right)}^{DG}\leq P_{\left(max\right)}^{DG}$$

### 3.8. Energy Participation between Grid and Prosumer

The grid net energy ($E_{n}^{G}$) traded with the utility in a single day is as follows: the energy import from the utility and the energy exchange to the utility are signified by the values of positive and negative of $P_{\left(t\right)}^{G}$, respectively.
(26)$$E_{n}^{G}=\sum_{t_{1}}^{t_{24}}P_{\left(t\right)}^{G}\times h$$

### 3.9. Stochastic Modelling of Solar PV

The generation of photovoltaic solar energy is very irregular and depends on the climate and output of solar irradiance. Under random conditions, the data of the whole year are analyzed. This article uses a solar irradiance model that has already been developed [49]. It also computes the parameters for the probability density (PDF) function of the standard normal distribution. By using Latin Hypercube (LHS) general sampling technique, 365 scenarios can be generated in 24 h. With the purpose of reduction in the calculation or computation burden, as mentioned in [50], a fast-forwarding technique is used to lessen the number of random scenarios generated up to almost 40 [50].
(27)$$F_{0}=\frac{1}{\sigma \sqrt{2\pi }}e^{-{\left(\frac{1-\mu }{2\sigma^{2}}\right)}^{2}}$$
(28)$$P_{\left(t\right)}^{\mathrm{PV}}=\eta_{\mathrm{PV}},{j\alpha }_{\mathrm{PV}}I$$

The normal distribution function [51] mentioned in Formula (27), is used to create an uncertainty model related to solar irradiance. Where $\eta_{PV}$; *j*; *I* and $\alpha_{PV}$ are the efficiency of the solar panel (17%), the solar irradiance pattern (kW/m^2)^, and the area of the solar panel (m^2^) respectively while the mean and standard deviation of the normal distribution are represented as *µ* and *σ*, respectively. Formula (28) shows the output solar power $P_{t}^{pv}$, which relies on the solar irradiance for an exact area. Figure 3 depicts the standard and mean deviation values of the photovoltaic irradiance predictable pattern for the Taxila geographical area, which includes the campus µG which has been considered. The Taxila region’s absolute location are “33.746° N” and “72.839° E”, respectively, corresponding to 5.3 kWh/m^2^/day [52].

### 3.10. Energy Participation between Grid and Wind Turbine

The wind power output $P_{\left(t\right)}$ traded with the utility grid is expressed in Formula (29) as:(29)$$P_{\left(t\right)}=\left\{\begin{array}{c} 0,\;V_{\left(t\right)}<V_{ci}\\ P_{r}^{WT}\times \left(\frac{\nu_{w}-\nu_{ci}}{\nu_{r}-\nu_{ci}}\right),v_{ci}<v_{\left(t\right)}<V^{r}\\ P_{r}^{WT}+\left(\frac{Y_{w}-V_{r}}{V_{ci}-V_{r}}\right)\times \left(P_{co}^{WT}-P_{r}^{WT}\right)\\ ,v_{r}<v_{w}<v_{co}\\ \;\;\;\;\;\;\;\;0,\;\;\;\;\;\;\;\;\;\;\;\;v_{co}<v_{w} \end{array}\right\}$$

The minimum cut-in speed required by the WT to generate power is expressed as ($\nu_{ci}$). The maximum cut-out speed at which maximum power is allowed to be generated is given as ($v_{co}$), if this speed is exceeded to avoid WT damage it has to be turned off.

### 3.11. Levelized Cost of Energy (LCOE)

In order to conduct a fair and an effective economic analysis of the system, the levelized energy cost is measured in multiple scenarios. It is denoted as the ratio between the entire system installation *cost* ($) and the energy produced (kWh). The *LCOE* for a storage or specific energy is expressed in $/kWh. It fulfills all related costs, consists of installation costs, operational costs, maintenance costs, and capital costs. It can also be observed as the minimal cost at which an electrical power should be sold during the life of the power generation or storage component to achieve balance or to attain the breakeven point [53]. Mathematically, the *LCOE* formula can be given as:(30)$$LCOE=\frac{Lifecycle\;cost\left(\$\right)}{Life\;time\;energy\;production\;\left(\mathrm{kWh}\right)}$$

### 3.12. Solution Methodology

Since the objective function and all related constraints of the proposed system model are basically linear models with many other integer variables, MILP programming is integrated to solve the linear optimization problem. The MILP technique is a common worldwide optimization method used to solve various kinds of optimization problems that are linked with marketing, scheduling, and optimal scheduling. Moreover, it has also been compared with many metaheuristic methods that provide suboptimal outcomes, but MILP provides the best optimal solutions and results. Hence, the MILP method is extensively used in EMS optimization [54]. The generic structure of MILP is given as follows:(31)$$\begin{array}{c} min\\ x \end{array}\;f^{t}x$$
(32)$$t_{0}\left\{\begin{array}{c} B.\;x\leq b\\ B_{eq}.x=b_{eq}\\ xb\leq x\leq yb \end{array}\right\}$$

In Formula (32), *xb*; *yb*; *x*; *b*; $b_{eq}$; and f are vectors, where $B_{eq}$; B is a matrix. In the initial stage, all the input data that are essential for the day are loaded one hour before each day arrives. Data include forecasted irradiance, forecasted temperature, load patterns, the ESS initial condition, TOU tariff information and its associated parameters. The simulation of the given optimization model is based on some regular interval prior to use every single hour. However, the proposed algorithm is basically simulated in MATLAB software, version-R2017a with Intel (R) Core (TM) i7-7700 @ 2.80 (GHz) processor with an 8 GB RAM.

## 4. Results and Discussion

The given model was implemented for the prosumer microgrid in Section 3 located in the Punjab province. The university has eight hostels, fourteen departments, and six faculties. At present, the university feeds its load from a 2 MW grid connection. The capacity of the campus rooftop PV installation is calculated to be 4 MW from a brief analysis of the area available for the campus rooftop.

According to the NEPRA (National Electric Power Regulatory Agency, Pakistan) which permits only 1 MW energy trading among the utility grids, we therefore have a limitation in installing 4 MW PV due to regulatory requirements and budget constraints. In our case, the distributed generation optimal sizing is also focused now compared to ref. [7], an onsite 2 MW solar PV installation is considered for comprehensive technical and economic analysis. Some other effects are also focused here to utilize the available diesel generator as a backup in case of power grid failure. Figure 4 shows the general flowchart diagram to control the proposed campus µG.

In addition, it is expected that the power grid has an efficient net-metering facility which allows power export regulation up to 1 MW to overcome the prosumer energy consumption cost. The campus load varies continuously because of the loads of hostel, academic blocks, administration offices, and housing colony inside the campus.

The implementation of solar PV in Pakistan is a feasible and workable solution to mitigate the energy crises; according to the report [55], Pakistan, producing 5100 kWh solar energy from 1 MW solar plant per day. Thus, we devised a solution in this work, 320 sunny days/year and 9 sunlight hours/day. Furthermore, a BESS system was also considered for our approach. By implementing a PV system for the campus µG, lithium-ion batteries are proposed with the advantages of their long lifetime, superior efficiency, healthy energy density, high reliability, and low self-discharge.

### 4.1. Case Study

In this case study, an optimum scheduling of microgrid is introduced for all seasons, spring, summer, autumn, and winter in Pakistan. Variations in load patterns are observed typically for all seasons and for the ease in analysis both the patterns are considered the same for the seasons respectively.

In Pakistan, some months just like January to August are the peak energy consumption months, as these are considered to be the peak load months. However, peak load patterns for all months are taken for economic analysis while considering worst-case scenarios. To analyze the economic benefits, energy generated from PV can be exported to the utility to gain maximum benefits. The actual energy consumption of the campus is considered for typical days and to analyze the electrical energy cost on a regular basis, the data is taken from the local substation meters.

The loads of admin and academic blocks are set to be high when the campus is open, while the peak demands of energy in the hostels and the residential colony are observed till midnight. Table 1 signifies multiple parameters that are interlinked with the system, whereas electricity price information on the TOU scheme is described in Table 2. The detailed data of solar irradiance used here are taken from [56], and the data characteristics are modeled and analyzed using the probability distribution function (PDF) that has already been mentioned in Equation (19). The main objective of PDF is to produce the solar irradiance pattern regularly on a daily basis, while the solar irradiance pattern generated earlier estimates the PV generation output power using Equation (20).

### 4.2. Multiple Case Scenarios

In the case study, the multiple case scenario is discussed in which exchange investigation and energy consumption with the help of price-based data are also stated. Multiple scenarios have been devised here to understand the energy demand for all the seasons as shown in Figure 5.

Scenario 1(a): In the first scenario, the power demand for the campus is provided completely from the grid. No PV, wind, ESS, or the diesel generator are available here for the campus. With the help of the time of use (TOU) tariff, the operational cost of energy is found out to be $1430.8. The LCOE is calculated here as 0.101 $/kWh, in this case. The outcome of the results indicate that the energy day-to-day operational cost is extremely high in the first scenario and this will be used as a case study for comparing and analyzing with other case scenarios for the summer season.

Scenario 1(b): In the second scenario, the solar PV is combined with the prosumer microgrid, and is integrated to export the surplus energy to the utility as well as feeding the load, as shown in Figure 6. The solar PV generates 8623.7 kWh that signifies the efficacy of PV in the summer season. The LCOE taken for the solar PV is 0.055 $/kWh here. However, the electricity net cost per day is reduced by 42.9% which becomes $758.5 from the base value.

Scenario 1(c): In the third scenario, ESS is integrated with the PV and utility connection, as shown in Figure 7. The proposed approach is implemented to find the net energy cost of $734.9 obtained and to schedule the battery charging/discharging patterns optimally with the consideration of all associated components of costs in this third scenario. The LCOE was calculated to be $0.056/kWh with the help of the TOU based tariff while considering BESS optimal scheduling as mentioned in Table 3.

The minor increment in the LCOE is because of the BESS cost involved here in this scenario. The comparison with the base scenario 1(a) reveals that it reduced the net cost of electricity about 41.2%. The energy trade with the utility grid is also indicated in Figure 6 in which +ve and −ve values signify the energy import and export. The ESS optimal scheduling result shows that the battery end operation at the same amount of SOC, i.e., operates at 50% exactly according to the day beginning. Moreover, the ESS wisely saves the surplus energy in off-peak and peak hours, it discharges accordingly to reduce the operational cost of energy.

Scenario 1(d): In the fourth case, the campus microgrid integrates the diesel generator (DGen) with solar PV and the BESS system to reduce the peak consumption power from the grid from (7:00 PM to 11:00 PM). The grid imports energy of maximum up to 50 kW which is the limit for the grid connection and the limit for the output power of the DGen is set at 600 kW through these peak hours, as seen in Figure 8.

After BESS optimal scheduling, the electricity net cost was found out to be $680.6 per day. The LCOE calculated, in such a case, is 0.058 $/kWh which is then compared with the base scenario 1(a), it was found that it is 37.2% less with 3.517600 s execution/computational time.

Scenario 1(e): In the proposed scenario, the wind turbine system (100 kW) is incorporated with ESS, solar PV, diesel generator, and the grid connection, as shown in Figure 9. The wind power and wind speeds tend to be (3–5) times higher in the months of March and April. The campus µG considered the rated power of wind to be 25 kW while considering various factors in which hub height is 36.6 m, rated speed of the wind is 14 m/s, wind cut-in speed is 3.5 m/s, and wind cut-out speed is 25 m/s is the better choice amongst the wind turbines considered.

The LCOE calculated the wind to be 0.060 $/kWh which is 36.6% less compared to the 31% [7]. This is analyzed with the integration of the WT system with ESS, solar PV, diesel generator and the grid connection, 5.6% increased saving in the electricity net cost for the UET Taxila campus.

### 4.3. Sizing and Carbon Emissions Effects on Energy Cost

The effects of multiple sizing of solar PV incorporation in prosumer µG on the cost of purchasing energy from the grid and the reduction of CO^2^ emissions per day were analyzed. When solar integration doubled, GHG emissions also were minimized by two times as well as for the cost reduction.

The bar graph in Figure 10 also illustrates the various types of PV integration in the given model and their impact on the electricity cost purchased from the utility. Based on the obtained values in the above-mentioned cases, we can examine the difference in the operating costs of energy.

The analysis demonstrates that integrating distributed generation systems has numerous benefits, including self-consumption, load flexibility, and cost reduction. As a result, the proposed method can be integrated to minimize the operating cost of campus electricity consumption. It basically requires a control facility to optimally control all the types of sources and loads. In addition, offloading the grid also improves grid efficiency through the integration of renewable energies. Capital and installation costs will be allocated in some cases, which will incentivize campus stakeholders to put more money in battery installation and DG.

### 4.4. Economic Effects on Optimal BESS Sizing

The analysis also demonstrates the system components power ratings, capital costs, replacement costs, operation and maintenance costs, payback period, system efficiency, and their lifetimes.

The objective components include solar PV, BESS, converter, wind turbine, DGs, grid, and other extra equipment that is included in the microgrid system. The effects of various system components that are already running in the system are analyzed. The system NPV cost is analyzed with different components combined in the system. It was also analyzed that the system with 2000 kW solar PV, 600 kW DG, 6000 kWh Li-ion battery capacity, and 1200 rating converter has a 13.1 M $ net present cost for the system. If the system utilized components with ratings 3000 kW solar PV, 700 kW generator, 7000 kW BESS, and 1400 kW converter, then the net present cost analyzed is 15.6 M $ approximately as shown in Figure 11.

However, a BESS solution was selected for the U.E.T Taxila in the microgrid system as an optimal solution of 8925.7 kW of PV, 600 kW of DG, 25 kW of WT, and with 5000 kW BESS system for summer, and vice versa. An optimal system was adopted for the campus microgrid by calculating 668.8 $/kWh as a daily electricity cost which is an optimal solution. However, the savings analyzed with the previous case scenarios [7] gave 36.3% savings analyzed comparatively 5.3% higher than the previous case scenario of 31%, as shown in Table 4. It was analyzed that with the optimal BESS solution with the incorporation of the WT system, savings are analyzed comparatively higher which is the best possible solution for the university campus microgrid.

## 5. Conclusions

In this study, an optimal scheduling of ESS and the impacts of solar PV were studied on a campus µG to lessen the energy operating costs for a commercial prosumer with the help of real load data. The proposed system considered solar PV, battery storage systems, and diesel generators under multiple scenarios, and analyzed their effects in various conditions. The optimal scheduling problem was simulated in MATLAB and modelled as a mixed integer linear problem taking battery life into account. The TOU based tariff (price based-DR) was studied here and ESS employed as a flexible DR system that could be charged and discharged intelligently at various timings to fulfil the objective of cost reduction without affecting durability. Without ESS or DG, all the required energy of the campus µG is provided by the utility company, resulting in increased operating costs. However, when PV, WT, DGen, and ESS are incorporated with the prosumer µG, the percentages of daily savings are 36.3%, respectively. The environmental impact of multiple sizes of PV installations was also studied here, and it was determined that the system with 2000 kW solar PV, 600 kW DG, 6000 kWh Li-ion battery capacity, and 1200 rating converter has a 13.1 M $ net present cost for the system. The reduction in the cost of electricity depends on various parameters such as demand, feed in tariffs (FIT), locations, etc. In Pakistan, the FIT has similar costs of buying and purchasing of electricity similar to those in many countries while the cost of selling the electricity to the utility is comparatively less than the cost of buying the electricity from the utility. As a result, investors can expect their electricity costs to increase by 20–30% by investing in on-site PV and ESS systems on an optimal schedule based on FIT, location, and level of load consumption. This leads to the conclusion that the optimum charge–discharge strategy for ESS plays an important part in the economic performance of prosumer buildings with internal RER installations. DG uncertainty, with more complex mathematical models with several storage systems considering DR types as well as a sensitivity analysis will be studied in future work.

## Figures and Tables

**Figure 1 sensors-22-06150-f001:**
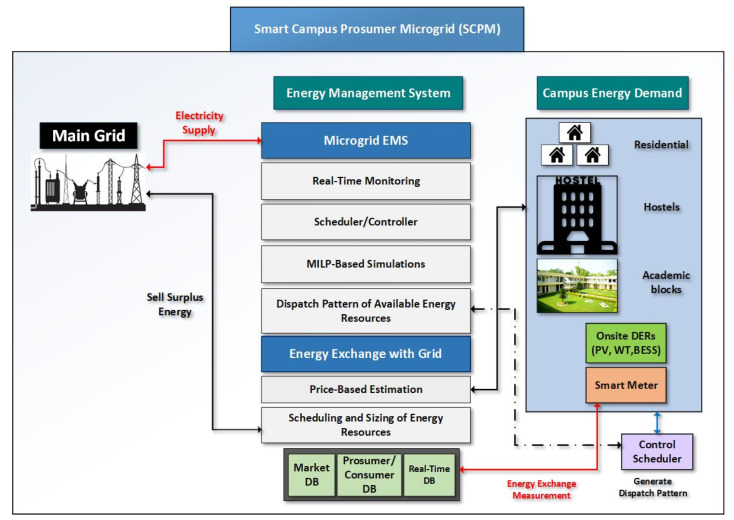
Proposed Conceptual model of EMS.

**Figure 2 sensors-22-06150-f002:**
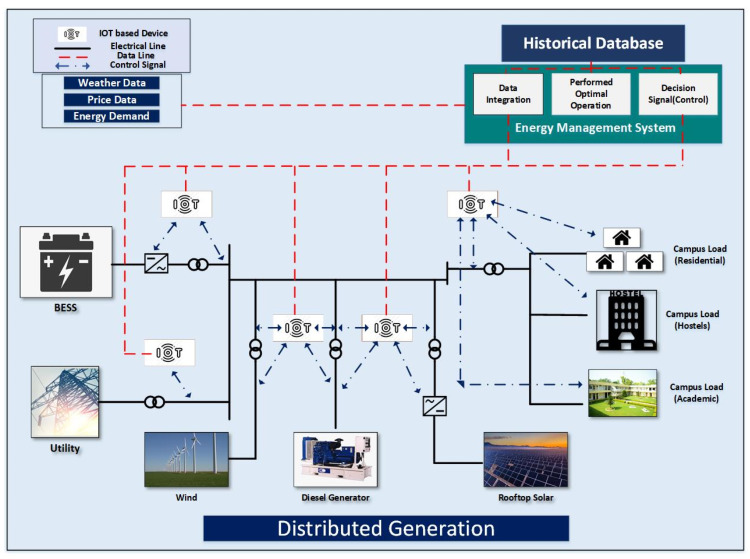
Proposed architecture of the system.

**Figure 3 sensors-22-06150-f003:**
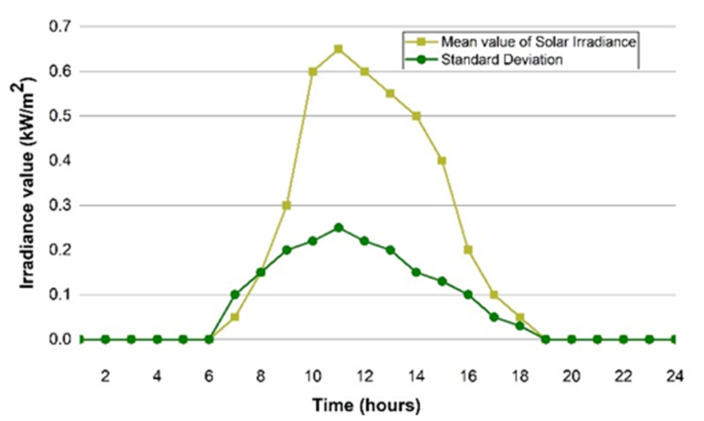
Standard deviation and mean value curves.

**Figure 4 sensors-22-06150-f004:**
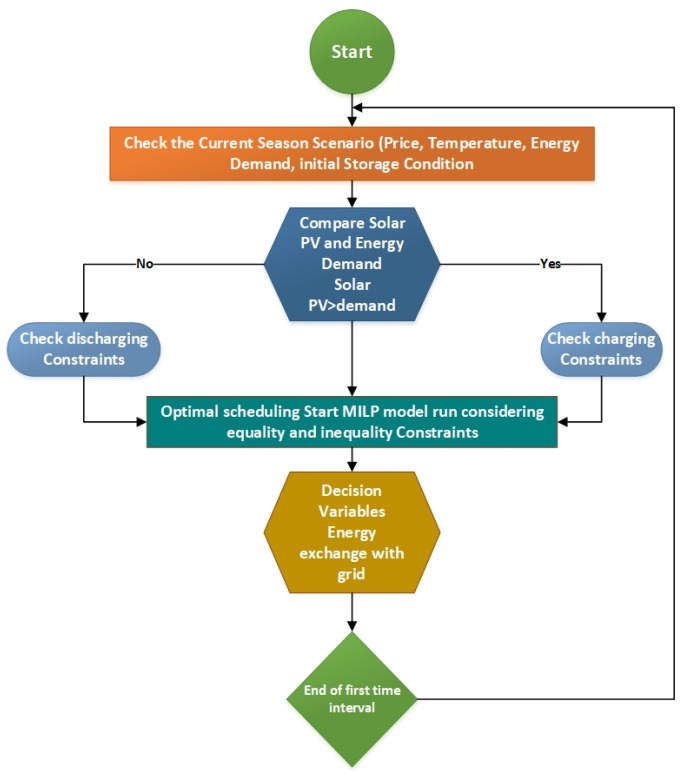
Proposed methodology of the given solution.

**Figure 5 sensors-22-06150-f005:**
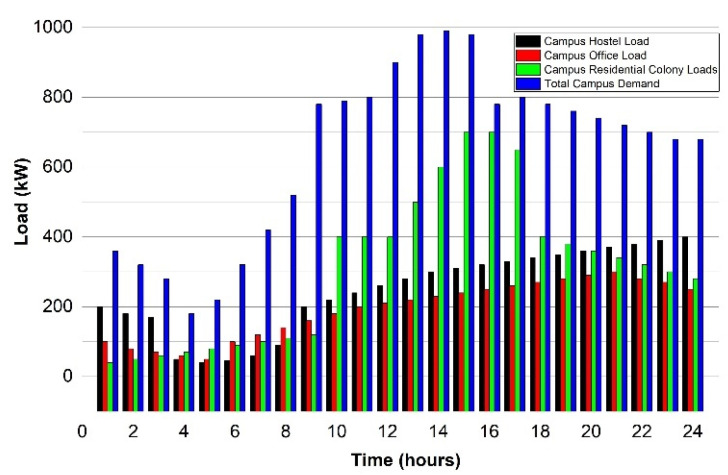
Load pattern behavior of campus.

**Figure 6 sensors-22-06150-f006:**
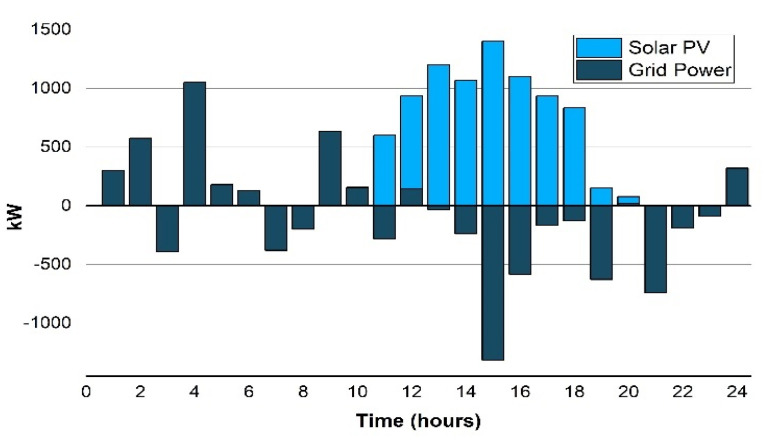
Scenario 1(b): Energy exchange with the power grid.

**Figure 7 sensors-22-06150-f007:**
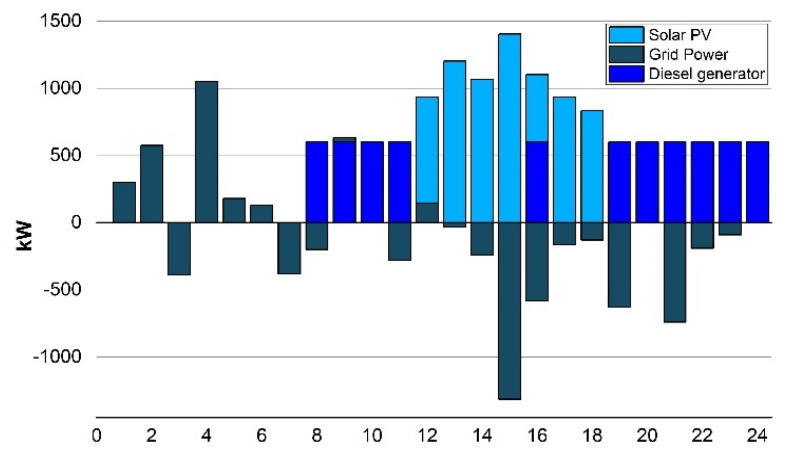
Scenario 1(c): Energy exchange with the grid.

**Figure 8 sensors-22-06150-f008:**
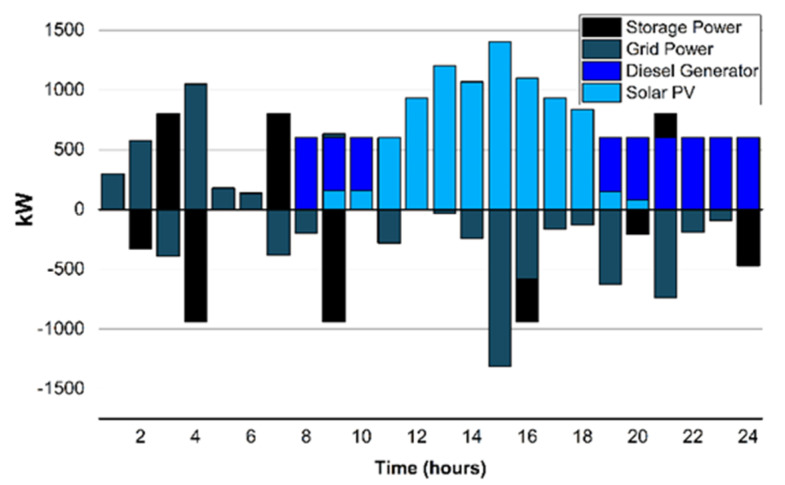
Scenario 1(d): Energy exchange with the grid.

**Figure 9 sensors-22-06150-f009:**
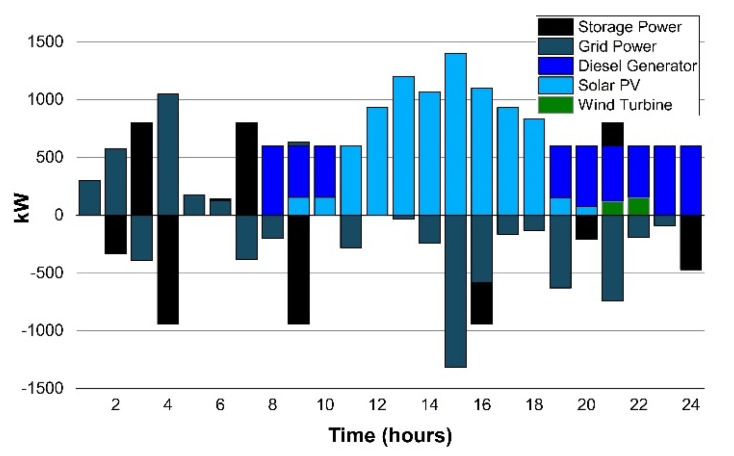
Scenario 1(e): Energy exchange with the grid.

**Figure 10 sensors-22-06150-f010:**
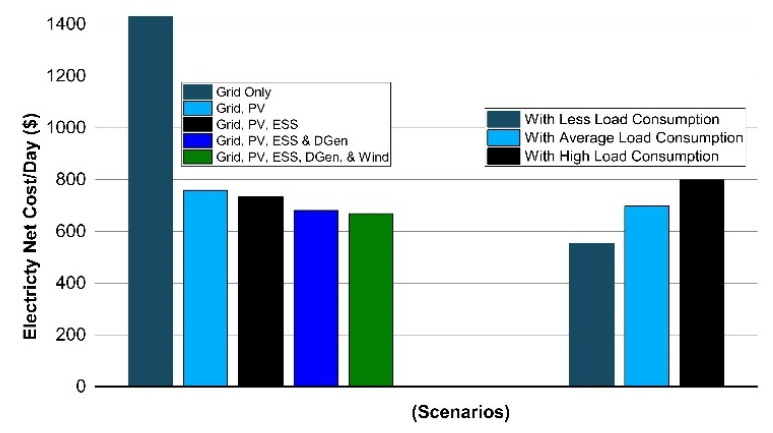
Electricity net cost analysis in multiple scenarios.

**Figure 11 sensors-22-06150-f011:**
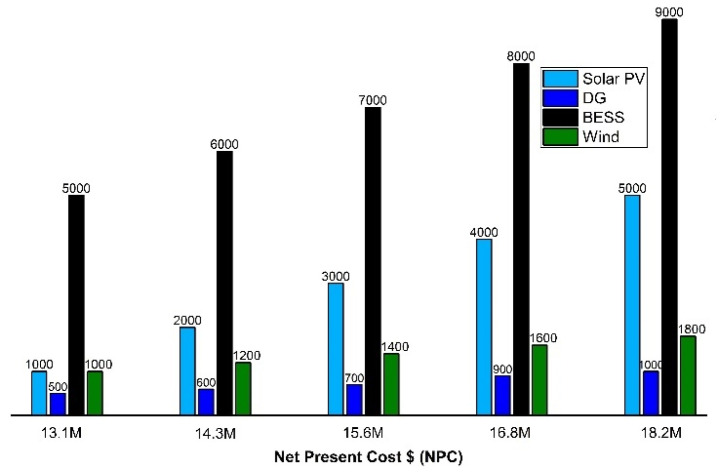
Net present cost for different components used in the system.

**Table 1 sensors-22-06150-t001:** Optimal Sizing System Parameters.

Parameters	Value	Parameters	Value
$P_{rated}^{PV}$	2000 kW	$C^{ES}$	800 kWh
$P_{\left(t,\max\right)}^{G}$	2000 kW	$P_{\left(t,\min\right)}^{G}$	−1000 kW
$P_{\left(t,\max\right)}^{bat}$	800 kW	$P_{\left(t,\min\right)}^{bat}$	−800 kW
$BSOC_{\left(\max\right)}^{b}$	90%	$DOD_{\left(\max\right)}^{b}$	0.95
$BSOC_{\left(\min\right)}^{b}$	50%	Battery Lifetime (LTY)	10
BESS Fixed-price	70.875	SOHM	0.6

**Table 2 sensors-22-06150-t002:** Electricity price distribution per unit.

Tariff Pricing	
Timing (Hours)	Unit Prices ($)
12:00 a.m. to 7:00 p.m.	0.10
7:00 p.m. to 11:00 p.m.	0.138
11:00 p.m. to 12:00 a.m.	0.10

**Table 3 sensors-22-06150-t003:** Multiple case results.

Cases	Only Grid	PV	ESS	DG	Wind	Energy Import by Grid (kWh /Day)	Electricity Generated from Prosumer (kWh /Day)	Grid Electricity Net Cost/Day ($) *	CC ** ($/ Day)	Electricity Net Cost without CC/Day ($) ^1^	Electricity Net Cost CC/Day ($)	LCOE ($/kWh)	% Saving
									A	B	(C = B − A)		
(a)	✓	×	×	×	×	141,72.5	-	1430.8	-	1430.8	1430.8	0.101	-
(b)	✓	✓	×	×	×	5548.8	8623.7	610.7	165	923.5	758.5	0.055	42.9
(c)	✓	✓	✓	×	×	5548.8	8623.7	711.5	165	899.9	734.9	0.056	41.2
(d)	✓	✓	✓	✓	×	4784.2	8623.7	768.2	155	835.5	680.6	0.058	37.4
(e)	✓	✓	✓	✓	✓	4643.2	8893.9	546.4	145	813.8	668.8	0.060	36.6

* This only covers the cost of grid power, not the costs of additional components such as PV, ESS, WT, and/or DGen. ^1^ The LCOE for every scenario is used to calculate this cost. PV’s LCOE is estimated to be 0.048 $/kWh [7]. The cost of installation of DGen and ESS are partially compensated by contributing 0.15 $/kWh and 0.06 $/kWh, correspondingly, to the suggested provided model, which already includes the O&M costs of ESS and/or DGen in different cases [7]. ** If the prosumer is listed within the carbon development mechanism (CDM), he/she will receive a carbon credit (CC) [7].

**Table 4 sensors-22-06150-t004:** Comparison of the existing works compared with the proposed method.

Ref.	Year	Application	Technique	Remarks	Savings
[23]	2018	Campus µG	MILP	ESS Degradation cost, Peak demand	5.32%
[57]	2018	Residential Level	MILP	Frequency regulation	7%
[28]	2019	Residential µG	LP	Grid outage	16%
[58]	2020	Campus µG	MILNP	Electrical peak mitigation	23%
[59]	2021	Campus µG	MILP	ESS Degradation cost, Peak demand	5.27%
Proposed Model	2022	Campus µG	MILP	Self-consumption, ESS Degradation, Demand response, Optimal sizing & Economic analysis	36.3%

## Data Availability

Not applicable.

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
