# Peer review of "Scheduling and Sizing of Campus Microgrid Considering Demand Response and Economic Analysis"

_sensors, 2022, doi:10.3390/s22166150_

Round 1

Reviewer 1 Report

1) The introduction should be improved and better highlight the scope of the paper as well as its objectives towards the gaps in the literature

2) The literature review should be focused on the campus microgrid concept

3) The data elaboration in the case study should be improved by better highlighting the expected load and the consideration regarding renewable energy production

4) The legislative framework of the country should be taken into consideration

5) The novelty of the proposed optimization method is not clear. Even if the authors report improved savings with regards to other methods, the constraints of the system are pretty common. Please clarify, both in the introduction, the case study as well as the results the components of the proposed method that improve the method's results.

6) Figures are not well compiled into pdf. Please rectify this.

7) English improvements are needed at different sections of the article

8) The insights that the proposed case study bring to the campus microgrid design and operation problem should be clarified to improve the article's academic value

Author Response

Response to Reviewer 1 Comments

Point 1: The introduction should be improved and better highlight the scope of the paper as well as its objectives towards the gaps in the literature

Response 1: Thank you very much for your effort to improve the quality of our paper. Yes, the introduction section needs improvement, so, we have improved the introduction section while focusing on better highlighting the objective and scope of the research in a more appropriate way. These changes have been highlighted with “Yellow”.

Point 2: The literature review should be focused on the campus microgrid concept

Response 2: Thank you for pointing it out. Yes, some parts of the literature review were not focused for the campus microgrid, it was focused as conventional microgrids, so, we have revised it all from campus mircrogrids’ point of view. These changes have been highlighted with “Yellow”.

Point 3: The data elaboration in the case study should be improved by better highlighting the expected load and the consideration regarding renewable energy production.

Response 3: Expected load has already been mentioned in the table 3 as the campus daily energy import grid is 14172.5 (kWh/day), also, regarding with RER generations per day, which is already mentioned there.

Point 4: The legislative framework of the country should be taken into consideration.

Response 4: Yes, the legislative framework has also been considered, as the NEPRA allows 1-2 MW of energy exchange among grids, so, all of the related frameworks have also been focused on here in this study. More, it has already been discussed in the results and discussion section.

Point 5: The novelty of the proposed optimization method is not clear. Even if the authors report improved savings with regards to other methods, the constraints of the system are pretty common. Please clarify, both in the introduction, the case study as well as the results the components of the proposed method that improve the method's results.

Response 5: Thank you again for your valuable suggestion. Yes, the novelty of the proposed methods is not clear, we have clarified in the introduction section as well as in the conclusion to highlight the novelty of this paper as suggested by our honorable reviewer. These changes have been highlighted with “Yellow”.

Point 6: Figures are not well compiled into pdf. Please rectify this.

Response 6: Yes, the figures have not been compiled well. So, we have again compiled it in a proper format as suggested by our honorable reviewer. It can be checked.

Point 7: English improvements are needed at different sections of the article

Response 7: Thank you, honorable reviewer, for this suggestion, Yes, the English quality is somehow poor, so, we have proofread the whole paper, and carefully improve the English level in these sections.

Point 8: The insights that the proposed case study bring to the campus microgrid design and operation problem should be clarified to improve the article's academic value

Response 8: Thank you again for your valuable suggestion. Yes, this is needed to further clarify our proposed case study, so, it has been improved in the case study section as well as we have gone through both the introduction section and lastly conclusion section to further clarify the proposed case study for our campus microgrid system.

Thank you very much for your valuable suggestions to improve the quality of our manuscript.

……………Thank You……………

Reviewer 2 Report

The paper shows interesting results but it has shortcomings needing major revision. Please clarify the comments below 

1) Introduction should be rewritten to reduce the number of paragraphs. Discussion on each reference was put in one paragraph and this fault led to a high number of paragraphs. Authors should group a number of references with the same scope, advantages or disadvantages.

2) Please point out major disadvantages of previous studies by using bullets and then state your solutions to overcome these points. 

3) Please clarify the novelties of the paper and then show the contributions in brief.

4) The reviewer found similar papers with the work. So, please clarify the differences between your work and others in Introduction. 

- Optimal Operation of the Campus Microgrid considering the Resource Uncertainty and Demand Response Schemes

-Optimal scheduling for campus prosumer microgrid considering price based demand response 

- An Optimal Scheduling and Planning of Campus Microgrid Based on Demand Response and Battery Lifetime

- Multiobjective Optimization Model considering Demand Response and Uncertainty of Generation Side of Microgrid

-Generation-Load Coordinative Scheduling considering the Demand-Response Uncertainty of Inverter Air Conditioners  

- Sustainable Solutions for Advanced Energy Management System of Campus Microgrids: Model Opportunities and Future Challenges 

5) Please point out the improvement of your method as compared to original one in theory by using equations and figure. Then, pointing out the improvement in numerical result by referring to (Finding optimal solutions for reaching maximum power energy of hydroelectric plants in cascaded systems).

6) Authors compared their results with others in Table 4. But I see the work/systems in the paper and others are much different. How could you compare these studies each other? Please clarify the systems applied these studies. 

7) Please add Figure to describe the considered system in numerical results.

8) Please show and explain your data. Also, add optimal solution of your method and others for comparisons. 

9) Authors should discuss the difficulties as well as advantages of the work.

9) Please make comparison between your method and its original method by using figures for the solution process

Author Response

Response to Reviewer 2 Comments

Point 1: The introduction should be rewritten to reduce the number of paragraphs. Discussion on each reference was put in one paragraph and this fault led to a high number of paragraphs. Authors should group a number of references with the same scope, advantages or disadvantages.

Response 1: Thank you very much honorable reviewer for your effort to improve the quality of our paper. Yes, the introduction section needs improvement, so, we have reduced the number of paragraphs and improved the introduction section while focusing on grouping the same type of referencing and sections together in a more appropriate way. Much changes have also been suggested by another reviewer, and these changes have been highlighted with colors such as “Green”.

Point 2: Please point out major disadvantages of previous studies by using bullets and then state your solutions to overcome these points.

Response 2: Yes, it is not explained briefly. We have added a brief explanation above the main contribution of the paper at page 6. Thank you for giving this suggestion. It is also highlighted in “Green”.

Point 3: Please clarify the novelties of the paper and then show the contributions in brief.

Response 3: Thank you very much for your valuable suggestion. Yes, the novelty of the paper is not clear, so, we have discussed it in detail in the introduction section as well as in the conclusion to highlight the contribution of this paper as suggested honorable reviewer. These changes have been highlighted with “Green”.

Point 4: The reviewer found similar papers with the work. So, please clarify the differences between your work and others in Introduction.

- Optimal Operation of the Campus Microgrid considering the Resource Uncertainty and Demand Response Schemes

-Optimal scheduling for campus prosumer microgrid considering price based demand response

- An Optimal Scheduling and Planning of Campus Microgrid Based on Demand Response and Battery Lifetime

- Multi objective Optimization Model considering Demand Response and Uncertainty of Generation Side of Microgrid

-Generation-Load Coordinative Scheduling considering the Demand-Response Uncertainty of Inverter Air Conditioners 

- Sustainable Solutions for Advanced Energy Management System of Campus Microgrids: Model Opportunities and Future Challenges.

Response 4: Yes, these studies related to the same campus microgrids aspect, but it has used different methodologies with different systems. Our work utilized MILP (Mixed Integer Linear Programming) methodology, with optimal scheduling and sizing of the campus microgrids with economical analysis.

For example: -

  1. In – (Optimal Operation of the Campus Microgrid considering the Resource Uncertainty and Demand Response Schemes), the author utilized mixed-integer nonlinear programming (MINLP) with two pricing schemes (TOU and RTP) and also Resource Uncertainty has been focused. But, they didn’t mention the net present cost as well as, their proposed model cost has been much higher at that time.
  2. In – (Optimal scheduling for campus prosumer microgrid considering price based demand response), the author used didn’t focus on the ecnomical analysis and optimal sizing of the campus microgrid, that is why their cost is comparatively higher than our proposed model.
  3. In – (An Optimal Scheduling and Planning of Campus Microgrid Based on Demand Response and Battery Lifetime), the author utlized Quadratic Programming as their method, and we used MILP approach, author used RTP with TOU tarrif, also their system only have DG, PV, and grid, our proposed system has DG, PV, Wind, Grid, and ESS, so our proposed system has high energy consumption from renewable energy resources.
  4. In – (Multi objective Optimization Model considering Demand Response and Uncertainty of Generation Side of Microgrid), the author used multiobjective scheduling optimization model and they have different energy systems such as Gas Turbine, WPP, etc. So, it has a different system not comparable with our microgrid system.
  5. In – (Generation-Load Coordinative Scheduling considering the Demand-Response Uncertainty of Inverter Air Conditioners), the author studied the Demand-Response Uncertainty of Inverter Air Conditioners and we studied the Scheduling and Sizing of Campus Microgrid.
  6. In – (Sustainable Solutions for Advanced Energy Management System of Campus Microgrids: Model Opportunities and Future Challenges), as it is a “survey paper”, so, the author reviewed multiple campus microgrids which have smart and innovative energy systems to present better options for the university campuses.

Point 5: Please point out the improvement of your method as compared to original one in theory by using equations and figure. Then, pointing out the improvement in numerical result by referring to (Finding optimal solutions for reaching maximum power energy of hydroelectric plants in cascaded systems).

Response 5: Thank you for pointing it out. Yes, this section is not compared in the results and discussion section, so, we have mentioned the referred papers according to the aforementioned results. The sections which have been improved can also be seen in the mentioned papers. These changes have been highlighted with “Green”.

Point 6: Authors compared their results with others in Table 4. But I see the work/systems in the paper and others are much different. How could you compare these studies each other? Please clarify the systems applied these studies.

Response 6: Yes, the microgrids which is compared has the same nature of problems (Electrical Peak Mitigation, ESS Degradation Cost, Peak Load Demand) as main focus is to reduce the operational costs, or to reduce their energy consumption cost, irrespective of their systems. It has performed different case studies to reduce their energy consumption costs, that is comparatively lead to same problem in end as how much saving, it has been calculated in that scenario, as some have a total 11.5M $ investment cost, while some invested 10.5M $, depends upon the labor and installation cost, as this table is compared generally. As, the focus is on the savings for the particular microgrid, not just the systems utilized of the microgrid, keeping in view the net present cost of the microgrids, which is somehow the same. If that system is incorporated into our proposed system, it will not give as much benefit as our proposed system.

Point 7: Please clarify the novelties of the paper and then show the contributions in brief.

Response 7: Yes, as discussed above, we have mentioned it in detail in the introduction section as well as in the other sections to highlight the contribution of this paper as suggested honorable reviewer. These changes have been highlighted with “Green”.

Point 8: Please show and explain your data. Also, add optimal solution of your method and others for comparisons.

Response 8: Yes, a thorough explanation has been added both in the results and discussion part, as it is compared comparatively with the other studies also above table 4. These changes have been highlighted with “Green”.

Point 9: Authors should discuss the difficulties as well as advantages of the work.

Response 9: Thank you for your valuable suggestion. Yes, this section has been added above the literature review which has also been suggested by another reviewer.

Point 10: Please make comparison between your method and its original method by using figures for the solution process.

Response 10: Yes, the method has already been compared in results and discussion section. Most of the solution process that has been implemented for the U.E.T Taxila microgrid, has already been upgraded and in improved form, especially if figure 2 has been considered, then, it has 2 new systems BESS, and Wind, which was not incorporated before, in the recent literatures.

Thank you very much for your valuable suggestions to improve the quality of our manuscript.

……………Thank You……………

Round 2

Reviewer 1 Report

Thanks for addressing my comments 

Author Response

Thank you very much for improving the quality of our paper. 

Reviewer 2 Report

Dear authors,

You have addressed approximately all comments and suggestion of the reviewer. So, your paper is much improved. However, The current contributions has many paragraphs, which should be shorten. Please group references with the same topic in the same paragraphs. 

Author Response

The paper has been updated as suggested by our honorable reviewer.

Once again thank you very much for your effort to improve the quality of our manuscript.